# Predictive Factors for Poor Outcome following Chemonucleolysis with Condoliase in Lumbar Disc Herniation

**DOI:** 10.3390/medicina58121868

**Published:** 2022-12-18

**Authors:** Shu Takeuchi, Junya Hanakita, Toshiyuki Takahashi, Tomoo Inoue, Manabu Minami, Izumi Suda, Sho Nakamura, Ryo Kanematsu

**Affiliations:** 1Department of Spinal Disorders Center, Fujieda Heisei Memorial Hospital, Shizuoka 426-8662, Japan; heisei.spine-jh@ny.tokai.or.jp (J.H.); heisei.t-taka@ny.tokai.or.jp (T.T.); emuminami@gmail.com (M.M.); izumis306@gmail.com (I.S.); nsho.1018@gmail.com (S.N.); ryo.knmt@gmail.com (R.K.); 2Department of Neurosurgery, Shizuoka General Hospital, Shizuoka 420-8527, Japan; 3Department of Neurosurgery, Saitama Red Cross Hospital, Saitama 330-0081, Japan; tomoo49@gmail.com

**Keywords:** lumbar disc herniation, chemonucleolysis, condoliase, chondroitin sulfate ABC endolyase, calcification, ossification

## Abstract

*Background and Objectives:* Condoliase, a chondroitin sulfate ABC endolyase, is a novel and minimally invasive chemonucleolytic drug for lumbar disc herniation. Despite the growing number of treatments for lumbar disc herniation, the predicting factors for poor outcomes following treatment remain unclear. The aim of this study was to determine the predictive factors for unsuccessful clinical outcome following condoliase therapy. *Material and Methods:* We performed a retrospective single-center analysis of 101 patients who underwent chemonucleolysis with condoliase from January 2019 to December 2021. Patients were divided into good outcome (i.e., favorable outcome) and poor outcome (i.e., requiring additional surgical treatment) groups. Patient demographics and imaging findings were collected. Clinical outcomes were evaluated using the numerical rating scale and Japanese Orthopaedic Association scores at baseline and at 1- and 3-month follow-up. Pretreatment indicators for additional surgery were compared between the 2 groups. *Results:* There was a significant difference in baseline leg numbness between the good outcome and poor outcome groups (6.27 ± 1.90 vs. 4.42 ± 2.90, respectively; *p* = 0.033). Of the 101 included patients, 32 received a preoperative computed tomography scan. In those patients, the presence of calcification or ossification in disc hernia occurred more often in the poor outcome group (61.5% vs. 5.3%, respectively; *p* < 0.001; odds ratio = 22.242; *p* = 0.014). Receiver-operating characteristics curve analysis for accompanying calcification or ossification showed an area under the curve of 0.858 (95% confidence interval, 0.715–1.000; *p* = 0.001). *Conclusions:* Calcified or ossified disc herniation may be useful predictors of unsuccessful treatment in patients with condoliase administration.

## 1. Introduction

Lumbar disc herniation is one of the most common disorders in patients with lumbosacral radiculopathy, affecting 40% of the population during their lifetime [1,2]. Although 60–80% of these patients experience spontaneous relief of their symptoms within 6–12 weeks [3], lumbar disc herniation often impairs quality of life [4,5]. Therefore, effective management of this disorder is very important.

The majority of patients with lumbar disc herniation typically start with conservative treatments, including pharmacologic therapy, physical therapy, and epidural anesthetic injection [6]. Surgical intervention is considered when a satisfactory outcome is not obtained. In recent decades, treatments for lumbar disc herniation have become less invasive, and minimally invasive surgical techniques such as percutaneous endoscopic lumbar discectomy have been developed [7]. However, these treatments require skilled surgeons and have risks of complications including dural tears, nerve root injury, and post-operative infection [8]. 

Chemonucleolysis is a safe and less invasive treatment for lumbar disc herniation that was first reported in 1964 [9]. This procedure involves the injection of an enzyme into the intervertebral disc to dissolve the disc component, resulting in reduced intradiscal pressure on the nerve root. Chymopapain and collagenase are the two main enzymes used clinically. However, chymopapain was confirmed to cause rare but severe adverse side effects such as fatal anaphylactic reactions and was discontinued in 2002 [10,11,12]. 

Condoliase (chondroitin sulfate ABC endolyase) is a safer and more effective chemonucleolysis enzyme that was originally developed in Japan and launched in August 2018 [13,14]. The safety and efficacy of condoliase were previously reported, with almost 80% of patients having a favorable outcome [14,15]. The enzyme selectively degrades chondroitin sulfate and hyaluronic acid chains contained in proteoglycans, which are rich in nucleus pulposus. That makes it possible to reduce water content and pressure in herniated nucleus pulposus. Therefore, the effects of condoliase therapy may depend on the degree of water content in the nucleus pulposus. However, as yet, no definitive predictive factors associated with unsuccessful condoliase treatment have been reported. Thus, the aim of the present study was to determine the predictive factors related to poor clinical outcome of chemonucleolysis with condoliase in patients with lumbar disc herniation.

## 2. Materials and Methods

### 2.1. Patient Selection and Data Collection

Between January 2019 and December 2021, 106 patients with lumbar disc herniation underwent chemonucleolysis with condoliase at our institution. The diagnosis of lumbar disc herniation was based on meticulous neurological examination for lumbosacral radiculopathy, and was confirmed by magnetic resonance imaging (MRI). Chemonucleolysis was performed in patients who wished to avoid surgery and had refractory leg symptoms or back pain despite the use of pharmacological treatment or periradicular infiltration for more than 1 month. Of these patients, 5 were excluded because of loss to follow-up (4 patients) or a requirement for lumbar surgery at a different level within 2 weeks (1 patient). Thus, a total of 101 patients were retrospectively analyzed. To determine factors relating to ineffective treatment, patients were divided into good outcome (G-group) or poor outcome (requirement for additional surgical treatment; P-group) groups (Figure 1). Patient demographics including age, sex, body mass index, smoking, medical history, morbidity period, and follow-up period were recorded from medical records.

### 2.2. Treatment Procedure

Patients were placed in the prone position. Under local anesthesia, a 22-gauge disc puncture needle was inserted into the center of the nucleus pulposus with fluoroscopic guidance. After confirming the needle tip position, condoliase (1.25 U/1.0 mL saline; ^®^HERNICORE; Seikagaku Corporation, Tokyo, Japan) was injected by a registered board-certified spinal surgeon. All patients were hospitalized on the day of intervention and discharged the next day [16].

### 2.3. Clinical and Imaging Evaluations

Clinical outcomes were assessed using the numerical rating scale (NRS) and the Japanese Orthopaedic Association (JOA) score. Post-operative NRS and JOA scores were evaluated at 1 month and 3 months after the procedure. For imaging evaluations, plain radiography and lumbar MRI were performed for all patients before the intervention. On MRI, the herniated disc location, Pfirrmann classification [17], Modic changes [18], the high-intensity zone (HIZ) of the protruded nucleus, and herniated disc migration were examined. On plain radiography, the rate of posterior intervertebral angle ≥ 5° and spondylolisthesis (defined as 3 mm vertebral slipping based on the lateral-lumbar radiograph) were assessed. Disc height was calculated with the following equation: disc height = (anterior disc height + posterior disc height)/(2 × upper adjacent vertebral body height) [14]. In accordance with a previous study, cephalad or caudal migration of >2 mm from the disc level was regarded as migration positive [15]. The HIZ of the protruded nucleus pulposus was defined as a brighter signal of the protruded nucleus pulposus than that of the non-protruded nucleus pulposus on sagittal T2-weighted MRI [19,20]. Of the 101 patients, 32 patients received a preoperative computed tomography (CT) scan, in which vacuum phenomena and the existence of calcification or ossification were evaluated.

### 2.4. Statistical Analysis

All statistical analyses were performed using statistical software (IBM SPSS v26.0; IBM, Chicago, IL, USA). The Mann–Whitney U test, chi-square test, and Fisher exact probability test were applied for univariate analysis. For multivariate analysis, logistic regression analysis was performed to calculate the odds ratios and 95% confidence intervals. Continuous data are expressed as the mean ± standard deviation. All *p*-values were 2-sided, and *p*-values < 0.05 were considered statistically significant. All patients had complete data.

## 3. Results

### 3.1. Clinical Evaluation

The baseline demographics and clinical characteristics of the patients are presented in Table 1. Of the 101 patients, 88 (87.1%) were included in the G-group and 13 (12.9%) were included in the P-group. There were no differences in age, sex, body mass index, morbidity period, or history of spine surgery at the same level between the 2 groups. There was a trend toward a higher baseline NRS scores of back pain and leg pain in the P-group. The mean NRS scores of leg numbness was significantly higher in the P-group compared with the G-group (6.27 ± 1.90 vs. 4.42 ± 2.99, respectively; *p* = 0.033). At both 1- and 3-month follow-up after chemonucleolysis, the NRS scores of back pain, leg pain, and leg numbness significantly improved in the G-group compared with the P-group (*p* < 0.01 for each) (Figure 2). Although an improvement in JOA scores was observed in both groups, this was only significant in the G-group (*p* < 0.01) (Figure 2). In the P-group, surgery was performed within 3 months due to the ineffectiveness of condoliase administration. All patients who underwent surgery got satisfactory outcome after surgery.

### 3.2. Imaging Evaluation

The imaging findings between the 2 groups are shown in Table 2. There were no differences in the classification of herniation, HIZ on protruded hernia, spondylolisthesis, Pfirrmann classification, or Modic change between the 2 groups. A posterior intervertebral angle ≥ 5° occurred more often in the P-group than in the G-group (2 [15.4%] vs. 1 [1.1%], respectively; *p* = 0.043). In the 32 patients who underwent CT scan prior to chemonucleolysis, calcified or ossified herniation significantly influenced the treatment outcome (1 [5.3%] vs. 8 [61.5%], respectively; *p* < 0.001) (Table 3). A representative case is shown in Figure 3.

Multivariate logistic regression analysis showed that the existence of calcification or ossification was a significant predictor of poor outcome for chemonucleolysis with condoliase (odds ratio, 22.242; *p* = 0.014) (Table 4). On the receiver-operating characteristics curve analysis of risk factors for unsuccessful treatment, the area under the curve was 0.858 (95% confidence interval, 0.715–1.000; *p* = 0.001) (Figure 4).

### 3.3. Adverse Events

Five patients in G-group experienced skin rash after administration of condoliase, which improved with oral antiallergic drugs within several days. Transient exacerbation of back or leg pain was observed in three patients (two in G-group and one in P-group). There were no severe adverse events such as anaphylactic reaction, neurological deterioration, and infection.

## 4. Discussion

The main finding of the present study was that calcified or ossified herniation was a risk factor for unsuccessful treatment with chemonucleolysis using condoliase. Previous studies demonstrated that 70–85% of patients achieved >50% improvement of NRS or visual analogue scale scores at 3 months after chemonucleolysis with condoliase [15,21,22,23,24]. The present study showed that >80% of patients were classified into the G-group (without requirement for additional surgery), and that the mean NRS scores of leg pain and leg numbness significantly improved by 63.9% and 64.4%, respectively, at 3 months, which was comparable with previous reports. Although the NRS scores of low back pain also significantly improved, this was by <50% (46.8% at 3 months). Because low back pain is multifactorial (e.g., myofascial pain, facet-mediated pain, and discogenic pain) [25], the effect may be relatively less than that for leg symptoms. 

Few studies have examined the negative factors associated with clinical outcomes of condoliase therapy. Banno et al. reported that patients with a history of discectomy, spondylolisthesis, or a posterior intervertebral angle of ≥5° had less improvement in symptoms after condoliase therapy [22]. Similarly, the present study showed significantly higher rates of patients with a posterior intervertebral angle of ≥5° in the P-group compared with the G-group (15.4% vs. 1.1%, respectively; *p* = 0.043). We also found a significant difference in preoperative leg numbness between the groups. In a retrospective study, numbness in patients with lumbar radiculopathy recovered much slower even after surgical decompression [26]. Thus, patients with sensory characteristics may have more unsatisfactory outcomes, and thus require additional surgery. 

CT scans are not routinely used as a diagnostic modality because of the potential cancer risk [27]. In our study, 32 patients underwent CT scans for surgical planning following severe spondylotic changes, CT myelography for detailed examination. Of the 13 patients who received a CT scan in the P-group, calcification or ossification was observed in 8 patients. The poor outcomes in patients with calcified or ossified lesions may relate to direct compression of the nerve root by the solid components themselves, leading to failure of condoliase treatment. Nevertheless, some of the calcified herniations in our patients were relatively small. Alternatively, several studies have reported a correlation between disc degeneration and calcification [28,29]. In the present study, classification according to Pfirrmann grading indicated disc degeneration in all patients (2 patients with grade 3 degeneration, 6 patients with grade 4 degeneration). The intervertebral disc is largely composed of cartilage made up of proteoglycan aggrecan and type II collagen [30]. The high proteoglycan content allows the nucleus pulposus to retain water. However, disc calcification reduces the water content [31]. Because condoliase acts on proteoglycans, changes in water content caused by calcification may result in ineffective treatment. In the present study, calcification or ossification in the disc hernia was an important predicting factor for unsuccessful treatment. Because MRI cannot clearly demonstrate the ossified or calcified changes, a preoperative CT examination in patients with disc degeneration is important for achieving satisfactory results of condoliase treatment for lumbar disc herniation.

### Limitations

This study has some limitations. First, the sample size was small, especially for the number of patients examined by CT scan. Second, because of the short follow-up period, the long-term outcomes for condoliase treatment were not assessed. Third, the retrospective design of this study may have introduced selection bias. Thus, further prospective studies with a large sample size are important to confirm our findings.

## 5. Conclusions

Calcification or ossification within disc hernia observed on CT scan is a useful predictive factor for unsuccessful treatment in patients with condoliase administration. 

## Figures and Tables

**Figure 1 medicina-58-01868-f001:**
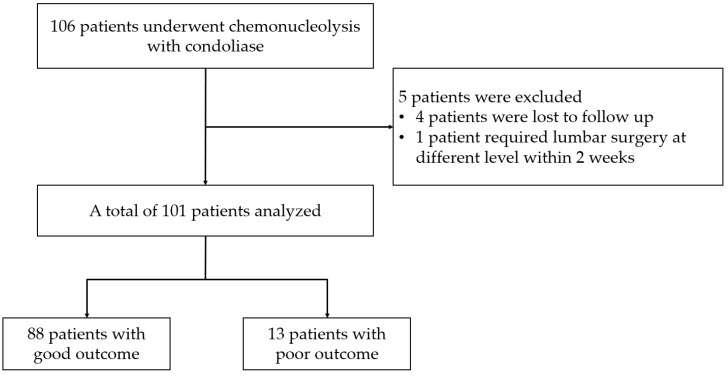
Patient selection flow.

**Figure 2 medicina-58-01868-f002:**
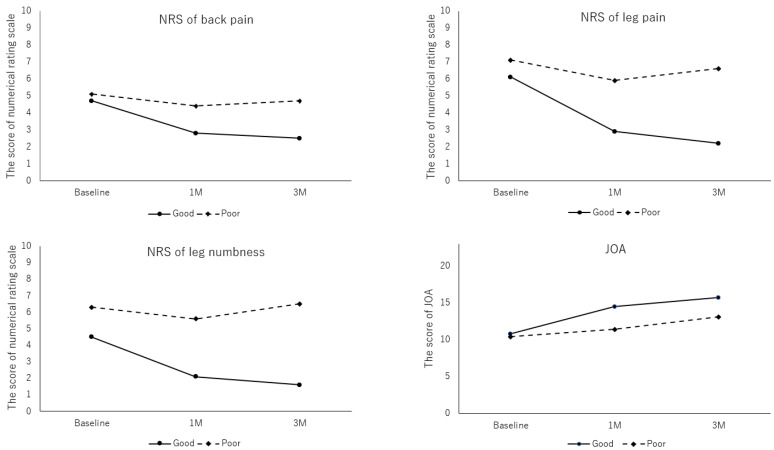
Changes in the numerical rating scale (NRS) of back pain, leg pain, and leg numbness, and the Japanese Orthopaedic Association (JOA) scores after chemonucleolysis with condoliase.

**Figure 3 medicina-58-01868-f003:**
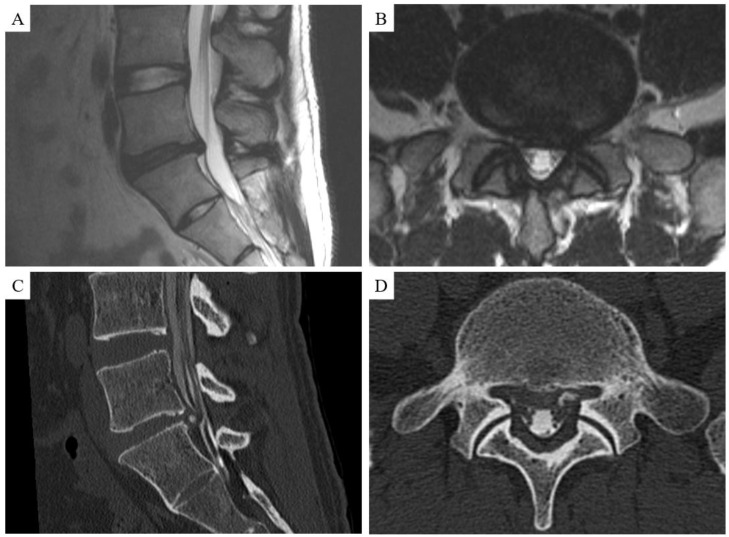
Illustrative example of a patient in the poor group with a calcified disc herniation. T2-weighted sagittal (**A**) and axial (**B**) magnetic resonance images show an L5/S1 disc herniation. Sagittal (**C**) and axial (**D**) computed tomography myelography helped to confirm the calcified disc herniation.

**Figure 4 medicina-58-01868-f004:**
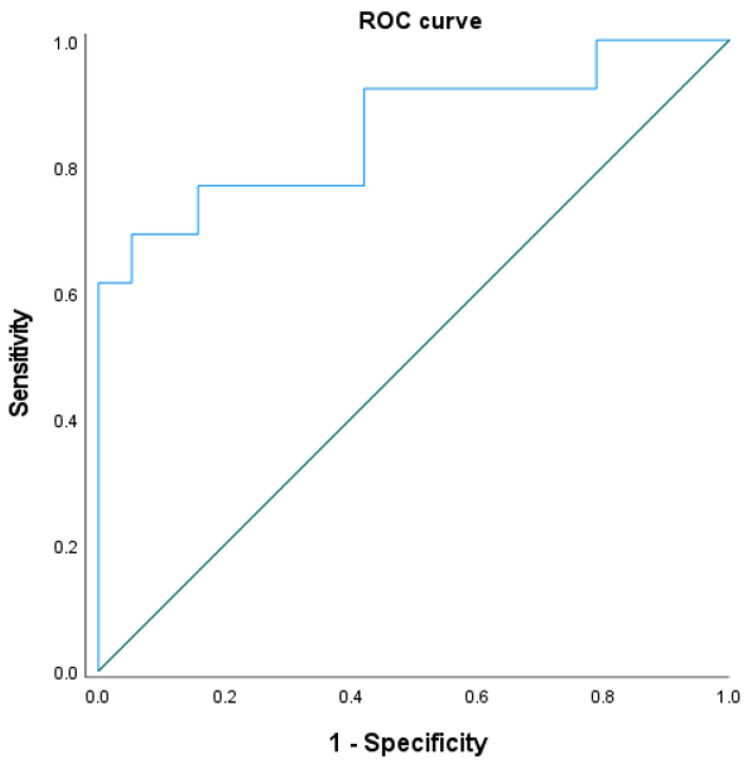
Receiver-operating characteristics (ROC) curve analysis of the logistic regression model provided an area under the curve of 0.858 (95% confidence interval, 0.715–1.000).

**Table 1 medicina-58-01868-t001:** Patient demographics.

Demographic	Total (n = 101)	Good Outcome (n = 88)	Poor Outcome (n = 13)	*p*-Value
Age	53.11 ± 18.68	53.55 ± 18.47	50.15 ± 20.55	0.594
Female	21 (20.8%)	20 (22.7%)	1 (7.7%)	0.292
BMI	23.61 ± 4.06	23.58 ± 4.14	23.81 ± 3.58	0.762
Smoking	58 (57.4%)	53 (60.2%)	5 (38.5%)	0.228
Morbidity period (months)	7.71 ± 11.40	8.09 ± 12.07	5.12 ± 4.17	0.604
History of spine surgery at the herniated level	17 (16.8%)	15 (17.0%)	2 (15.4%)	1.000
Preoperative JOA score	10.77 ± 4.01	10.82 ± 4.10	10.39 ± 3.48	0.783
Preoperative NRS				
Back pain	4.79 ± 2.38	4.75 ± 2.36	5.08 ± 2.53	0.595
Leg pain	6.27 ± 2.31	6.15 ± 2.39	7.08 ± 1.46	0.193
Leg numbness	4.66 ± 2.94	4.42 ± 2.99	6.27 ± 1.90	0.033

Values are presented as mean ± standard deviation or number (%). BMI, body mass index; JOA, Japanese Orthopaedic Association; NRS, numerical rating scale.

**Table 2 medicina-58-01868-t002:** Imaging characteristics.

Characteristic	Total (n = 101)	Good Outcome (n = 88)	Poor Outcome (n = 13)	*p*-Value
Foraminal herniation	5 (5.0%)	4 (4.5%)	1 (7.7%)	0.505
Extraforaminal herniation	4 (4.0%)	3 (3.4%)	1 (7.7%)	0.429
Cephalad/caudal migration	50 (49.5%)	43 (48.9%)	7 (53.8%)	0.775
HIZ on protruded disc	23 (22.8%)	22 (25.0%)	1 (7.7%)	0.288
Disc height	0.279 ± 0.077	0.278 ± 0.080	0.283 ± 0.053	0.756
Posterior intervertebral angle ≥ 5°	3 (3.0%)	1 (1.1%)	2 (15.4%)	0.043
Spondylolisthesis	5 (5.0%)	5 (5.7%)	0 (0%)	1.000
Herniation level				0.868
L1/2	1 (1.0%)	1 (1.1%)	0 (0%)	
L2/3	3 (3.0%)	3 (3.4%)	0 (0%)	
L3/4	12 (11.9%)	10 (11.4%)	2 (15.4%)	
L4/5	41 (40.6%)	35 (39.8%)	6 (46.2%)	
L5/S1	44 (43.6%)	39 (44.3%)	5 (38.5%)	
Pfirrmann classification				0.712
II	9 (8.9%)	8 (9.1%)	1 (7.7%)	
III	48 (47.5%)	43 (48.9%)	5 (38.5%)	
IV	41 (40.6%)	34 (38.6%)	7 (53.8%)	
V	3 (3.0%)	3 (3.4%)	0 (0%)	
Modic change				0.788
Type I	3 (3.0%)	3 (3.4%)	0 (0%)	
Type II	18 (17.8%)	16 (18.2%)	2 (15.4%)	
Type III	13 (12.9%)	12 (13.6%)	1 (7.7%)	

Values are presented as mean ± standard deviation or number (%). HIZ, high-intensity zone.

**Table 3 medicina-58-01868-t003:** Computed tomography findings.

	Total (n = 32)	Good Outcome (n = 19)	Poor Outcome (n =1 3)	*p*-Value
Vacuum phenomena	9 (28.1%)	7 (36.8%)	2 (15.4%)	0.249
Calcification or ossification	9 (28.1%)	1 (5.3%)	8 (61.5%)	<0.001

Values are presented as number (%).

**Table 4 medicina-58-01868-t004:** Logistic regression analysis for ineffective treatment.

Variable	β	SE	Wald	df	*p*-Value	Exp(β)	95% CI for Exp(β)	
							Lower	Upper
Age	−0.022	0.027	0.656	1	0.418	0.978	0.928	1.032
Preoperative NRS of leg numbness	0.35	0.211	2.741	1	0.098	1.419	0.938	2.148
Calcification or ossification	3.102	1.263	6.029	1	0.014	22.242	1.870	264.576

β, β coefficient; SE, standard error; Wald, Wald statistic; df, degrees of freedom; Exp(β), exponentiation of β coefficient; CI, confidence interval; NRS, numerical rating scale.

## Data Availability

Not applicable.

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
