# Peer review of "Predictive Factors for Poor Outcome following Chemonucleolysis with Condoliase in Lumbar Disc Herniation"

_medicina, 2022, doi:10.3390/medicina58121868_

Round 1

Reviewer 1 Report

General

The authors The aim of this study was to determine the predictive factors for unsuccessful clinical outcome following condoliase therapy

I really agree your concept and well written manuscript.

I am positive in terms of publication but still have several question for correction.

I have the following detailed comments

1.      Introduction

Well written. But as possible, hypothesis is located before purpose in the introduction.

2.     Methods

Well written

It is not specified which institution the target patients received the treatment at. For example, it may be a single-center study or a multi-center study.

In addition, it is not described whether the operator is the same person or, if not, how many. I don't think we can ignore the difference in skill level between the operators.

3.     Results

Well written

4.     Discussion

Well written

As your limitation, selection bias is most important problem in this manuscript. Even though this problem, there are no problem for publication.

5.     Conclusion

Well written

Author Response

Comment 1: 1. Introduction

Well written. But as possible, hypothesis is located before purpose in the introduction.

Authors’ response: We are grateful to Reviewer 1 for the positive remarks and a helpful suggestion. We have added hypothesis in the introduction line55-59.

Comment 2: 2. Methods

Well written. It is not specified which institution the target patients received the treatment at. For example, it may be a single-center study or a multi-center study.

In addition, it is not described whether the operator is the same person or, if not, how many. I don't think we can ignore the difference in skill level between the operators.

Authors’ response: We are grateful to Reviewer 1 for the positive remarks and a helpful suggestion. This is single center study. We have added “single center” in the Abstract  – Material and Methods line 15.

The operator is the same person. We have corrected it line83-84.

Reviewer 2 Report

The authors present a solid retrospective study of condoliase use in lumbar disc herniation. The design is reasonable but could be improved, the implementation is absolutely solid. 

If possible, I would recommend changing or at least discussing the following points:

1. lack of a control group. With a therapy method such as the condoliase application, it would make sense to evaluate a control group for comparison (e.g. purely conservative therapy). 

2. the authors rightly write in the introduction that in more than 50% of patients with a herniated disc the symptoms normalise spontaneously within a few weeks to months without any invasive therapy. It should somehow be discussed or supplemented why the patients treated here needed invasive therapy at all in the retrospective evaluation. 

3. from a personal point of view, I miss another detail in the evaluations. One of the most important characteristics that has emerged in recent years as essential for successful invasive therapy of a herniated disc and is being discussed more and more frequently scientifically is the duration of the radicular symptoms. In short, the longer the symptoms last, the less likely it is that invasive therapy will be successful. Here the authors could use the at least striking correlation of treatment failure with calcification of the herniated disc to discuss this point. It would be even better if the retrospective data could be supplemented by the duration of complaints at the time of therapy. 

Overall, I have to admit that the results are not really surprising or suggest new aspects, but I am not aware of a comparable publication - so I would support a publication if the above points were addressed. 

Author Response

Comment 1: lack of a control group. With a therapy method such as the condoliase application, it would make sense to evaluate a control group for comparison (e.g. purely conservative therapy). 

Authors’ response: We thank you very much for your suggestion. We agree with your comment, but the aim of this study is to evaluate the factors associated with ineffective condoliase treatment. Therefore, we didn’t include a control group this time. Regarding therapy method, we state in Materials and Methods – Treatment procedure.

Comment 2: the authors rightly write in the introduction that in more than 50% of patients with a herniated disc the symptoms normalise spontaneously within a few weeks to months without any invasive therapy. It should somehow be discussed or supplemented why the patients treated here needed invasive therapy at all in the retrospective evaluation. 

Authors’ response: Thank you for your comment. The reason for codoliase treatment is the failure of conservative treatment, pharmacological therapy and periradicular infiltration, for more than 1 month. We have inserted “for more than 1 month” line71

Comment 3:  from a personal point of view, I miss another detail in the evaluations. One of the most important characteristics that has emerged in recent years as essential for successful invasive therapy of a herniated disc and is being discussed more and more frequently scientifically is the duration of the radicular symptoms. In short, the longer the symptoms last, the less likely it is that invasive therapy will be successful. Here the authors could use the at least striking correlation of treatment failure with calcification of the herniated disc to discuss this point. It would be even better if the retrospective data could be supplemented by the duration of complaints at the time of therapy. 

Authors’ response: We appreciate your helpful suggestion. As you mentioned, longer duration of symptoms is associated with non-success in sciatica patients [Haugen, A.J.; Brox, J.I.; Grovle, L.; Keller, A.; Natvig, B.; Soldal, D.; Grotle, M. Prognostic factors for non-success in patients with sciatica and disc herniation. BMC Musculoskelet Disord. 2012, 13, 183.]. In fact, we also have evaluated the duration of symptoms (i.e. morbidity period) at the time of therapy, and there are no significant correlation of treatment failure in Table 1.

Round 2

Reviewer 1 Report

All the comments I asked last time were well presented in this version.